# DONUT-hole: DONUT Sparsification by Harnessing Knowledge and Optimizing Learning Efficiency

**Azhar Shaikh**[1]    **Michael Cochez**[1]    **Denis Diachkov**[2]    **Michiel de Rijcke**[2]    **Sahar Yousefi**[2*]

[1]Vrije Universiteit, Amsterdam, The Netherlands    [2]Sight Team, Prime Vision BV., Delft, The Netherlands

`{a.shaikh@student.,m.cochez@}vu.nl`
`{d.diachkov,m.d.rijcke,s.yousefi}@primevision.com`

## Abstract

This paper introduces DONUT-hole, a sparse OCR-free visual document understanding (VDU) model that addresses the limitations of its predecessor model, dubbed DONUT. The DONUT model, leveraging a transformer architecture, overcoming the challenges of separate optical character recognition (OCR) and visual semantic understanding (VSU) components. However, its deployment in production environments and edge devices is hindered by high memory and computational demands, particularly in large-scale request services. To overcome these challenges, we propose an optimization strategy based on knowledge distillation and model pruning. Our paradigm to produce DONUT-hole, reduces the model denisty by 54% while preserving performance. We also achieve a global representational similarity index between DONUT and DONUT-hole based on centered kernel alignment (CKA) metric of 0.79. Moreover, we evaluate the effectiveness of DONUT-hole in the document image key information extraction (KIE) task, highlighting its potential for developing more efficient VDU systems for logistic companies.

## 1   Introduction

Logistic companies handle a vast number of physical forms, invoices, and parcel labels that contain shipment details, as well as recipient and sender information. In order to efficiently manage the information provided by these documents, there is a high demand for a fast and reliable digitization process which significantly lowers the burden of manually tagging the documents. VDU systems are being developed to harness document image data to perform downstream tasks like document image classification, document visual question answering, document layout analysis, KIE that are critical in optimizing and automating various business processes across industries. Conventional VDU systems consist of three components 1) text detection 2) text recognition 3) VSU to establish relationship between elements. In recent years the text detection and recognition components have been merged into a deep learning based optical character recognition (OCR) system and the focus has been on improving the VSU component. To best of our knowledge, all the VSU can be categorized into graph based reasoning Yu et al. [2021], Liu et al. [2019], Sun et al. [2021] and attention based reasoning Xu et al. [2021, 2020], Hong et al. [2022]. Obtaining reliable results for the downstream tasks heavily relies on the critical cooperation of the VSU and the OCR components. Not only does separating VSU from OCR increase complexity, but it also results in longer computational time Kim et al. [2022]. To alleviate these shortcomings, Kim et al. Kim et al. [2022] introduced DONUT based on the state of the art transformer architecture Vaswani et al. [2017].

DONUT leverages attention mechanism to combine the textual and visual features together to make a direct mapping from the input image to a desired structured information format. Although the model performs efficiently, it has over two hundred million parameters which requires significant

Workshop on Advancing Neural Network Training at 37th Conference on Neural Information Processing Systems (WANT@NeurIPS 2023).

computational resources, making it unsuitable for (near) real-time productions that demand low latency. These large computational resources can incur steep financial as well as environmental costs. Alternatively training of small models has empirically been shown to be a hard optimization problem Ba and Caruana [2014] while also being limited in their expressivity.

To circumvent this issue, lately there has been significant interest in utilizing knowledge distillation Hinton et al. [2015] for model compression. The works in knowledge distillation can be classified into two distinct categories of task-oriented and task-agnostic approaches. Kim et al. proposed a sequence-level knowledge distillation for sequence-to-sequence language models Kim and Rush [2016]. In Xia et al. [2022] a task-oriented approach utilizing structured pruning has been introduced which jointly prune coarse- and fine-grained layers in the network, to effectively control the pruning decision of each parameter. To facilitate knowledge transfer from un-pruned to pruned models during optimization they suggested a layer-wise distillation strategy. In Chen et al. [2021] proposed a novel task-agnostic approach for sharing knowledge between teacher and student networks by utilizing connection path cross levels. Zafrir et al. presented an architecture-agnostic method for producing compressed up-stream language models using unstructured pruning Zafrir et al. [2021]. Huang et al. suggested a loss function based on correlation that explicitly captures the inherent relationships between different classes, as provided by the teacher Huang et al. [2022]. Also, they discussed that large gap between the student's and the teacher's size negatively impacts the knowledge distillation results. To alleviate this issue in Liu et al. [2022] an Embedding Assistant module was proposed, which aids to build a pseudo teacher assistant model by combining the student's transformer blocks and seamlessly bridge teacher and student models. Exploring the literature on knowledge distillation reveals the potential benefits of upstream pruning over downstream pruning Zafrir et al. [2021], Liang et al. [2023].

In this work, we aim to use the aforementioned methods to reduce the model density and size and consequently computational requirements of DONUT while preserving its performance for reading and document image KIE applications. We independently evaluate the role of knowledge distillation and model pruning on effectively sparsifying DONUT for the aim of model compression. We also show that a simple combination of pruning then distilling, dubbed prune-distill, is a very effective and efficient approach to compress DONUT. In addition to this we also outline the potential challenges and drawbacks of each of the approaches. Through rigorous experimentation and evaluation, we will determine the most effective approach for enhancing the performance and reducing the size of DONUT. For the first time in model distillation, we will employ the CKA Kornblith et al. [2019] measure to assess the representation similarity between the teacher and student networks for each training paradigm, inspired by Raghu et al. [2021]. In this work we achieved 54% reduction in model size, resulting in a sparse model suitable for utilization in resource-constrained applications.

## 2 Proposed method

### 2.1 Teacher Network

In all the experiments conducted in this study, we employ the DONUT architecture introduced by NAVIER Kim et al. [2022], dubbed DONUT-base, as the teacher network. The DONUT model has been trained on a synthetic data containing 500k samples for each language, including Chinese, Japanese, Korean, and English in addition to 11 million scanned English document images from the IIT-CDIP dataset Lewis et al. [2006]. We refer to this pretrained model as "DONUT-base-11M". Due to the unavailability of ground truth data for the IIT-CDIP dataset and our specific focus on the English language, we trained a scaled-down model with the same architecture on 500k synthetic English images to make fair comparisons. We refer to this model as "DONUT-base-0.5M."

### 2.2 Student Network

**DONUT-small:** During our initial attempts to compress the DONUT model, we identified the hidden dimension and layer depth of the Swin encoder, as well as the number of layers and the hidden and feedforward network dimension size of the decoder, as the primary bottlenecks affecting model size. We also observed that training a model with a transformer-based architecture from scratch can present challenges due to its memory and resource hungry behavior. With these constraints inplace we designed the student network by replacing the Swin-B encoder backbone with a pretrained Swin-T

Table 1: Models' configurations

| Model name | #non-embedding params | #embedding params | #total params |
|---|---|---|---|
| DONUT-base | 140M | 60M | 200M |
| DONUT-small | 37M | 29M | 66M |
| DONUT-base-pruned | 37M | 60M | 97M |

encoder backbone from the timm library provided by Hugging Face [1]. Additionally, we replaced the four-layer MBART decoder with a pretrained two-layer BART decoder, which is half the size of MBART, hosted on the Hugging Face model repository [2]. The selection of pre-trained visual encoders and textual decoders for designing the student network can result in challenges with cross-modal fusion due to the dimensions incompatibility of textual and visual tokens. To address this challenge, we introduce a novel approach in model distillation by utilizing an adapter bottleneck layer. Adapter bottleneck layer is a plug and play neural network component which serves to align the dimensions of textual and visual tokens, enabling effective cross-modal fusion Houlsby et al. [2019]. This layer acts as a bridge between the pre-trained visual encoder and textual decoder components, effectively aligning their dimensions. This modified model is referred to as "DONUT-small".

**DONUT-small-distilled:** KD reduces a model size with a minimal performance drop by distilling knowlege from a large model, dubbed teacher, to a small model, dubbed student. In this work, we deployed this strategy for performance boosting. For this goal we used DONUT-small as a student model and DONUT-base-11M as a teacher model. The resulted model from paradigm is named DONUT-small-distilled. Figure 2-left illustrates the DONUT-small-distilled.

**DONUT-base-pruned:** To deal with the issue of unavailability of smaller compatible checkpoints we proposed to first prune the teacher network "DONUT-base-11M" to a level of ∼50% of sparsity on the non-embedding parameters and use this pruned model as the student. For this goal we applied magnitude pruning which induces sparsity in neural networks by pruning weights based on their magnitudes. Weights are sorted by value and those below a certain threshold, depending on a predetermined sparsity level, are set to zero. This method effectively reduces the size and density, consequently increases the speed of the model. This pruned student model is referred to as "DONUT-base-pruned". Table 1 presents the model configurations along with their respective parameter sizes.

**DONUT-hole:** In the process of creating DONUT-hole, knowledge from the teacher model, i.e. DONUT-base-11M, is distilled to the student model, i.e. DONUT-base-pruned. This distillation involves applying the sparsity constraint used in Section 2.2, during gradient back-propagation to maintain sparsity during the distillation process. Figure 2-right illustrates the DONUT-hole.

## 3 Datasets and Metrics

### 3.1 Dataset

Following a similar approach as Kim et al. [2022], we assess the developed models in two phases: pre-training or upstream task, which focuses on reading, and fine-tuning or downstream task, which in this work aims at KIE. In this section we explain the dataset we use.

**Pre-training dataset:** For the pre-training phase of the model, we use Synthdog-EN dataset Kim et al. [2022]. Synthdog-EN has been generated using Synthetic Document Generator (SynthDoG) library from Wiki corpuses while background images are sampled from ImageNet Deng [2009]. The dataset consists of 500k training images and 500 validation images. To compensate the absence of test set in this dataset, we utilized Synthdog-EN configurations provided by DONUT library to further generate 791 images.

**Fine-tuning dataset:** To evaluate the performance of the compressed and sparse models further we test on the KIE task as a downstream goal. For this purpose we use two datasets of CORD-V2, published by NAVER Kim et al. [2022], and a commercial dataset collected by Prime Vision, called Parcel Reader. The CORD-V2 consists of 1K English receipt images splitted into 800, 100 and 100

---

[1]Hugging Face Model Repository: https://huggingface.co/models
[2]lucadiliello/bart-small

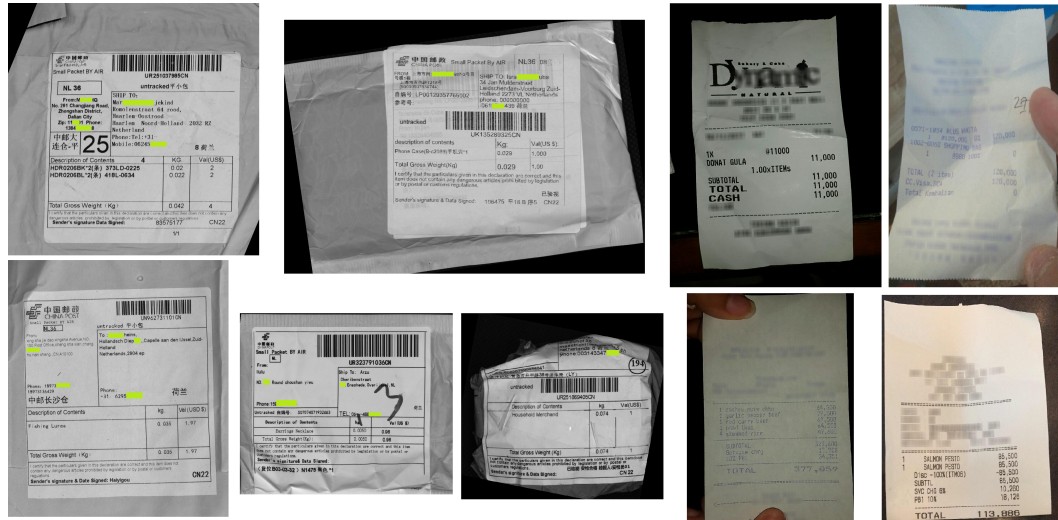

Figure 1: Samples of the downstream task datasets, left: Parcel reader, the personal details have been filtered out partially due to GDPR legislation (yellow regions), right: CORD-V2

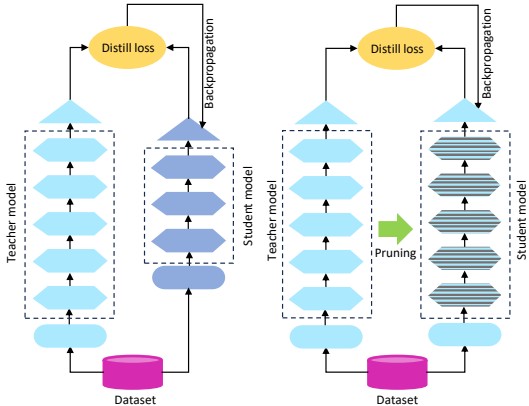

Figure 2: The proposed architecture, left: DONUT-small-distilled, right: DONUT-hole

images for train, validation and test sets, respectively. Parcel Reader dataset consists of the labels affixed to parcels encompassing the customs information such as item description, value, weight as well as receiver and sender details, such as name, address, phone number, etc. The dataset contains 1388 images splitted into 1109, 139 and 140 images for train, validation and test sets, respectively. For this data we extracted the English information from the images, as it was the language of interest for the customer, while also containing Chinese text. Figure 1 demonstrates some examples of the both mentioned datasets. Due to GDPR legislation we partially filtered out the personal details on the parcel labels.

## 3.2 Metrics

In this section we describe the metrics we use to evaluate the trained models.

**Normalized-Tree Edit Distance (N-TED)**   measures the similarity between two trees by calculating the minimum number of edit operations (such as insertions, deletions, and substitutions) required to transform one tree into another, and then normalizing it based on the maximum possible distance. N-TED accuracy is computed by taking the N-TED value between the predicted tree and the ground truth tree and subtracting it from 1. A higher N-TED accuracy indicates a closer match between the predicted and ground truth trees.

**Field F1** measures the harmonic mean of precision and recall for individual fields or labels. A higher Field F1 score indicates better performance in terms of both precision and recall for the respective field or label.

**Global representational similarity index** between two networks is computed by averaging the CKA layerwise similarity index scores. Let's consider two matrices, $X \in \mathbb{R}^{m \times d_1}$ and $Y \in \mathbb{R}^{m \times d_2}$, where $X$ represents the representations of a layer with dimensionality $d_1$, and $Y$ represents the representations of another layer with dimensionality $d_2$ neurons. The Gram matrices, $K = XX^T$ and $L = YY^T$, capture the pairwise similarities between examples.

To address the influence of mean effects, we introduce the centering matrix $H = I_n - \frac{1}{n}11^T$. By applying this matrix to $K$ and $L$, we obtain the centered similarity matrices $K_0 = HKH$ and $L_0 = HLH$, respectively. These centered matrices remove the column and row means, allowing for a more accurate assessment of similarity. The similarity between the centered similarity matrices is quantified using the Hilbert-Schmidt Independence Criterion (HSIC). To compute Gretton et al. [2005], the matrices are reshaped into vectors, and their dot product is calculated as:

$$\text{HSIC}_0(K, L) = \frac{\text{vec}(K_0) \cdot \text{vec}(L_0)}{(m-1)^2} \tag{1}$$

HSIC is sensitive to scaling variations in the original representations. To address this sensitivity, the Centered Kernel Alignment (CKA) normalizes HSIC, ensuring invariance to isotropic scaling. The CKA score between the similarity matrices $K$ and $L$ is obtained by dividing the HSIC score $\text{HSIC}_0(K, L)$ by the square root of the product of the HSIC scores $\text{HSIC}_0(K, K)$ and $\text{HSIC}_0(L, L)$:

$$\text{CKA}(K, L) = \frac{\text{HSIC}_0(K, L)}{\sqrt{\text{HSIC}_0(K, K) \cdot \text{HSIC}_0(L, L)}} \tag{2}$$

In summary, the CKA score provides a normalized measure of similarity between the similarity matrices $K$ and $L$ based on the HSIC scores. In this work, all the models have an equal number of encoder layers, ensuring consistent comparison at each encoder layer. However, there may be variations in the number of decoder layers among the models. To calculate the similarity index in such cases, the similarity of a network layer is summed with the corresponding layer in the teacher network.

Table 2: Results of the reading task on the SynthDog-EN dataset with input image resolution of 1280x960 and context length of 768 tokens

| Model | #Non-embedding Params | TED Accuracy |
|---|---|---|
| DONUT-base-0.5M | 140M | 0.92 |
| DONUT-small | 37M | 0.66 |
| DONUT-small-distilled | 37M | 0.83 |
| DONUT-base-pruned | 37M | 0.00 |
| DONUT-hole | 37M | 0.96 |

## 4 Experimental Results and Discussions

In this section, we discuss the experimental results for both downstream and upstream tasks. Additionally, we cover the representational similarity analysis.

### 4.1 Implementation Details

For both the reading and KIE tasks we use the Adam optimizer and follow the same learning rate schedules as in the DONUT paper. We use a batch size of 8, an input resolution of 1280x960 and maximum context length of 768 for all experiments. The DONUT-small type models are trained using a single Nvidia A100 and the DONUT-base type models are trained using two Nvidia A100 gpus.

Table 3: Results of the KIE task on the CORD-v2 dataset with input image resolution of 1280x960 and context length of 768 tokens

| Model | #Non-embedding Params | TED Accuracy | F1 Accuracy |
|---|---|---|---|
| DONUT-base-0.5M | 140M | 0.76 | 0.61 |
| DONUT-small | 37M | 0.55 | 0.37 |
| DONUT-small-distilled | 37M | 0.61 | 0.41 |
| DONUT-base-pruned | 37M | 0.00 | 0.00 |
| DONUT-hole | 37M | 0.85 | 0.75 |

Table 4: Results of the KIE task on the Parcel Reader dataset with input image resolution of 1280x960 and context length of 768 tokens

| Model | #Non-embedding Params | TED Accuracy | F1 Accuracy |
|---|---|---|---|
| DONUT-base-0.5M | 140M | 0.65 | 0.50 |
| DONUT-small | 37M | 0.44 | 0.26 |
| DONUT-small-distilled | 37M | 0.50 | 0.35 |
| DONUT-base-pruned | 37M | 0.24 | 0.05 |
| DONUT-hole | 37M | 0.73 | 0.57 |

## 4.2 Pretraining on the upstream task

The analysis of the reading task results on the SynthDog-EN dataset, as summarized in Table 2, reveals intriguing insights. Among student networks with comparable model capacity, the DONUT-hole stands out as the top performer, surpassing even the teacher network (DONUT-base). Conversely, the DONUT-pruned model exhibits the poorest performance, which can be attributed to the high level of sparsity resulting from the removal of a substantial proportion (∼50%) of network weights. Interestingly, the introduction of distillation after the pruning step shows a promising tendency to rejuvenate the network's performance. This observation suggests that the lower magnitude parameters removed during pruning are likely redundant and have limited impact on the network's overall performance. The distillation process seems to reestablish and reconnect previously broken connections within the network, essentially restoring its functional integrity. The effectiveness of distillation is further highlighted in the comparison between DONUT-small and DONUT-small trained with distillation, with the latter achieving an impressive 17% improvement. It is worth noting that, although the distillation-based approach does not surpass the performance of the teacher network in this specific case, it still yields significantly better results compared to the network trained without distillation. This demonstrates the potential of distillation as a valuable training strategy for enhancing student network performance. Moreover, the analysis of loss curves reveals an interesting observation. The prune-then-distill approach appears to offer computational efficiency, with the training process converging to a low training loss significantly faster compared to other student models. This suggests that the combination of pruning and distillation not only improves performance but also facilitates faster convergence during training. Overall, these findings shed light on the effectiveness of distillation in mitigating the performance degradation caused by pruning, underscoring its potential for network optimization and knowledge transfer in the context of student networks.

Figure 3 showed the scatter plots of N-TED values of DONUT-small DONUT-small-distilled and DONUT-hole vs DONUT-base-0.5M on the test dataset. Points closer to the diagonal line indicate instances where both models have similar predictions, while points far away from the diagonal suggest significant differences between the models. The scatter plot's dense clustering of points in the upper right region, close to one of the axes, suggests that the corresponding model consistently outperforms the other on the test set concerning the N-TED metric. Considering this fact, Figure 3a and Figure 3b show DONUT-base-0.5M outperforms DONUT-small and DONUT-small-distilled while Figure 3c illustrates DONUT-hole outperforms DONUT-base-0.5 in the upstream task.

## 4.3 Fine-tuning on the downstream task

Table 3 and 4 provide detailed insights into the performance of various models on the KIE task using the CORD-v2 and Parcel Reader datasets respectively. The DONUT-base model, with approxi-

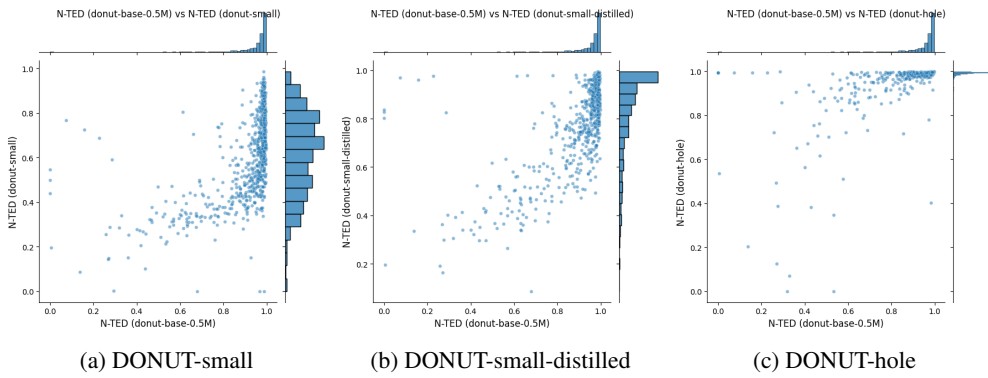

(a) DONUT-small      (b) DONUT-small-distilled      (c) DONUT-hole

Figure 3: Scatter plot of N-TED values of the trained models vs DONUT-base-0.5M on the SynthDog-EN test set

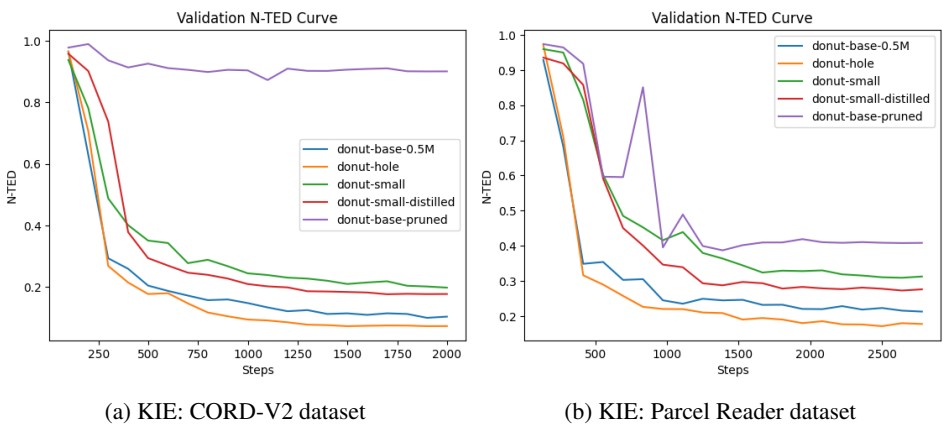

(a) KIE: CORD-V2 dataset      (b) KIE: Parcel Reader dataset

Figure 4: N-TED value on the downstream tasks

mately 140 million non-embedding parameters, performed reasonably well in the CORD-v2 dataset, outperforming the smaller DONUT-small model. The use of distillation significantly boosted the performance of the DONUT-small model, elevating both its TED and F1 accuracies. The DONUT-base-pruned model completely lost it's performance after pruning, having TED and F1 equal zero. Nevertheless, distillation managed to recover and even enhance its performance dramatically. On the Parcel Reader dataset, the DONUT-base model once again displayed better results compared to the DONUT-small model. Just as in the CORD-v2 dataset, distillation improved the performance of the DONUT-small model. The pruned model initially failed to deliver any substantial results, but with the application of distillation, its performance skyrocketed, surpassing all the other models. While the non-pruned models generally outperformed their smaller and pruned counterparts, the application of distillation significantly boosted the performance of the smaller and pruned models, sometimes surpassing their larger, non-pruned versions. Figures 4-a and 4-b illustrate the N-TED value during training for CORD-V2 and Parcel Reader validation sets. Both the figures indicate DONUT-hole is converging faster compare to it compartments in the downstream task.

## 4.4 Representational Similarity Analysis

In this section, we aim to share the results of our extensive study on representational similarity. We made use of CKA to contrast the representations across all the trained models against "DONUT-base-11M" network, taking into account both global and layer-specific similarities.

### 4.4.1 Global Similarity

The similarity between the trained networks and the reference "DONUT-base-11M" network was measured using aggregate CKA scores, as tabulated in Table 5. Generally, most networks exhibited

high similarity values, except for the "DONUT-base-pruned" model. The "DONUT-base-0.5M" network had the highest similarity to "DONUT-base-11M," which is expected since they have the same architecture but differ in training corpus size. Thus, these networks have acquired similar representations. However, the "DONUT-base-pruned" network showed a decrease in representational similarity due to the pruning process, which disrupts certain connections. Nevertheless, applying a distillation procedure to the pruned network improved its alignment with "DONUT-base-11M." It's worth noting that the achieved similarity is not as high as expected for a network initialized with a subset of weights from the original model. Despite this, the lower similarity value, combined with the model's good performance on reading and downstream tasks, suggests that the "DONUT-hole" model has likely learned a more compact and informative representation. Additionally, comparing "DONUT-small" to "DONUT-small" with distillation reveals that the distillation process effectively brings the representations of the two networks closer to each other. This demonstrates the potential of distillation in enhancing representation alignment.

It is worth noting that, the discrepancy between N-TED values tabulated in Table 2 and CKA values in Table 5, arises because N-TED focuses on the model's output performance, while CKA looks at the model's internal representations, which might not necessarily be indicative of its task-specific performance.

Table 5: CKA Similarity index when comparing representations with DONUT-base-11M

| Model | CKA similarity index |
|---|---|
| DONUT-base-0.5M | 0.93 |
| DONUT-small | 0.84 |
| DONUT-small-distilled | 0.90 |
| DONUT-base-pruned | 0.56 |
| DONUT-hole | 0.79 |

#### 4.4.2 Layerwise Similarity

Figure 5 provides insights into the learned representations of the networks, focusing on layer-wise similarity. Several important patterns and observations can be made from the analysis. Firstly, the enc.0 layer and dec.0 layer consistently exhibit high similarity with their counterparts in the "DONUT-base-11M" network. This suggests that these layers capture common and general features across all networks. Secondly, there is significant dissimilarity between the representations of the visual encoder and multimodal decoder layers. This indicates distinct processing functions within the network, with the visual encoder focusing on feature extraction and the multimodal decoder aligning visual and textual tokens. Furthermore, regardless of the variation in visual backbones (swin-T, swin-B, and pruned swin-B), the similarity scores within the visual encoder layers remain consistently high. This implies that the choice of visual backbone has limited influence on the learned representations, highlighting the robustness of the DONUT model in capturing essential visual features. The most substantial difference in representations is observed in the last decoder layer. Interestingly, the dec.0 and dec.1 layers in the "DONUT-base-11M" model show identical similarity values across all networks, indicating the same representation is learned for both layers. Comparing the decoder layers, the DONUT-hole in Figure 5e exhibits the greatest dissimilarity, particularly in the last decoder layer, suggesting a different token alignment approach in this scenario. Moreover, comparing Figures **??** and 5e demonstrates the benefits of distillation. Distillation improves the similarity of the last decoder layers' representations to those of the "DONUT-base-11M" network when pruning is not involved. This improvement in similarity is crucial for achieving good performance.

## 5   Conclusion

In this paper, for the first time, we explored compressing paradigms, such as prune, distill, and prune-distill, to reduce the density of the DONUT model. Distillation showed promising results, enhancing performance by 17% in the upstream task, i.e. reading. The introduction of distillation after pruning rejuvenated the network's performance and restored its functional integrity. The combination of pruning and distillation not only improved performance but also facilitated faster convergence

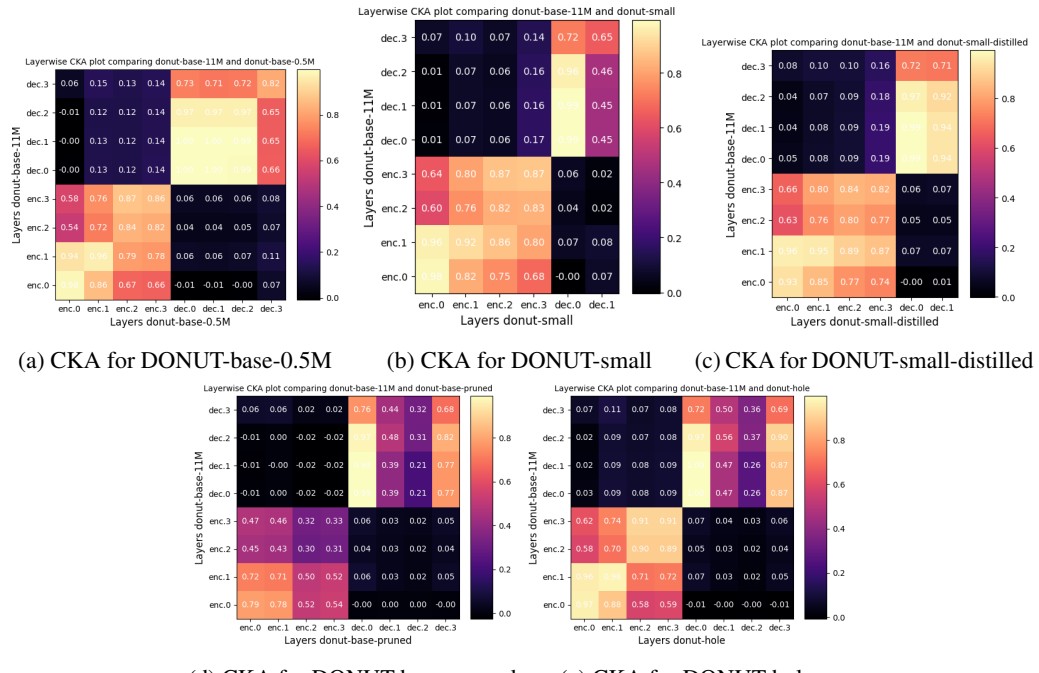

(a) CKA for DONUT-base-0.5M    (b) CKA for DONUT-small    (c) CKA for DONUT-small-distilled

(d) CKA for DONUT-base-pruned    (e) CKA for DONUT-hole

Figure 5: Visualizing Layerwise CKA Representational Similarity Index Heatmaps comparing representations of the trained models and DONUT-base-11M

during training, as concluded from both the comparison metrics and the representational similarity analysis presented in the paper. We named our proposed model DONUT-hole. DONUT-hole is a sparse model, reducing the DONUT size by 54%, while preserving performance as effectively as DONUT. We also evaluated the trained models for the upstream tasks on the SynthDog-EN dataset, and a downstream task, i.e., KIE on both CORD-V2 and a commercial dataset called Parcel Reader. Results showed DONUT-hole outperforms all the models trained by the other compressing paradigms. The findings highlight the effectiveness of distillation in enhancing student network performance and optimizing network knowledge transfer.

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
