# OpenReview forum: "DONUT-hole: DONUT Sparsification by Harnessing Knowledge and Optimizing Learning Efficiency"
_NeurIPS.cc/2023/Workshop/WANT — WANT@NeurIPS 2023 Poster_

### Official Review · Reviewer_sfj9 · 2023-10-22
**Good Industrial Application Paper on Knowledge Distillation for OCR**

**Confidence:** 4

**Review:**

The paper addresses the problem of compressing large dense vision-language models like DONUT for more efficient deployment. This is an interesting real-world industrial use-case study.

The authors explore a new architecture in DONUT-small, albeit it is not really effective. Distillation after pruning helps recover performance, showing it mitigates degradation from pruning. The prune-then-distill approach is  effective. The student model reduces parameters by 54% while maintaining accuracy competitive with the original DONUT model. CKA similarity analysis provides insights into how representations change with different training approaches. Results on document image reading and key information extraction tasks demonstrate applicability to real document understanding problems.

Issues:

* The distillation, pruning and combined approaches are standard techniques.
* More ablation studies isolating the impact of specific design choices would strengthen the methodology.
* Additional efficiency metrics like inference time, memory usage would help quantify real-world gains.
* More analysis of tradeoffs between accuracy, compression rate and efficiency would be useful.
* Testing on other architectures besides DONUT could better showcase generality.

An accept is recommended as the paper is an interesting industrial application paper, showing how computational benefits arise from careful pruning and distillation for an industrial use-case.

---

### Official Review · Reviewer_EyK9 · 2023-10-25
**Valuable work for practical applications but no scientific novelty**

**Confidence:** 4

**Review:**

Summary
This paper proposes to apply pruning and knowledge distillation techniques for DONUT model compression. The obtained compressed model named DONUT-hole achieves approximately 50% compression ratio in terms of total number of parameters while shows good performance in several evaluations. An adapter bottleneck layer used in DONUT-small model helps to align visual encoder and textual decoder components.
Despite this is the first work on DONUT compression to the best of my knowledge, this paper applied quite straightforwardly well-known existing methods, pruning and KD, without any improvements. The validity of comparing of DONUT-base-0.5M, trained on quite small dataset, with DONUT-hole, distilled from well-trained original model DONUT-base-11M, is arguable (Tables 2, 3, 4, 5).

Quality: the proposed models are evaluated on several datasets, but it's difficult to make confident conclusions on their efficiency due to issues mentioned above.

Novelty: the paper introduces new modified compressed versions of DONUT model but doesn't propose new compression methods or techniques.

Clarity: the paper is well-written, well-organized and easy to follow.

Significance: compression of large well-performing VDU models is an important task for industry.

Pros:
* Clear structure, well-written.
* Interesting result for real-world applications.

Cons:
* Weakly supported conclusions
* No scientific novelty

---

### Meta-Review · Area_Chair_Rf7A · 2023-10-26

**Recommendation:** Accept (Poster)
**Confidence:** 4

**Metareview:**

While pointing out some notable issues, the reviewers are generally positive about this work. The AC agrees that despite the limited technical contribution, the pipeline put together has its merit. The authors are strongly encouraged to address the issues pointed out by the reviewers in the updated manuscript.

---

### Decision · Program_Chairs · 2023-10-28

**Decision:**

Accept (Poster)

**Comment:**

We thank the authors for their time and contribution to WANT and we are pleased to share that after the reviewing process the paper has been accepted. Congratulations! We encourage the authors to consider reviewers' feedback for the improvement of the camera-ready version. We hope to see you in person at the workshop and brainstorm on efficient training research together!